# Comparative Evaluation of Artificial Neural Networks and Data Analysis in Predicting Liposome Size in a Periodic Disturbance Micromixer

**DOI:** 10.3390/mi12101164

**Published:** 2021-09-28

**Authors:** Ixchel Ocampo, Rubén R. López, Sergio Camacho-León, Vahé Nerguizian, Ion Stiharu

**Affiliations:** 1Tecnologico de Monterrey, School of Engineering and Sciences, Av. Eugenio Garza Sada 2501 Sur, Monterrey 64849, Mexico; sergio.camacho@tec.mx; 2Departments of Oncology & Pathology, Research Institute of the McGill University Health Centre, 1001 Décarie, Montreal, QC H4A 3J1, Canada; ruben.lopezsalazar@mail.mcgill.ca; 3Department of Electrical Engineering, École de Technologie Supérieure, 1100 Notre Dame West, Montreal, QC H3C 1K3, Canada; vahe.nerguizian@etsmtl.ca; 4Department of Mechanical and Industrial Engineering, Concordia University, 1455 de Maisonneuve Blvd. West, Montreal, QC H3G 1M8, Canada

**Keywords:** artificial neural networks, micromixer, liposome, data analysis

## Abstract

Artificial neural networks (ANN) and data analysis (DA) are powerful tools for supporting decision-making. They are employed in diverse fields, and one of them is nanotechnology; for example, in predicting silver nanoparticles size. To our knowledge, we are the first to use ANN to predict liposome size (LZ). Liposomes are lipid nanoparticles used in different biomedical applications that can be produced in Dean-Forces-based microdevices such as the Periodic Disturbance Micromixer (PDM). In this work, ANN and DA techniques are used to build a LZ prediction model by using the most relevant variables in a PDM, the Flow Rate Radio (FRR), and the Total Flow Rate (TFR), and the temperature, solvents, and concentrations were kept constant. The ANN was designed in MATLAB and fed data from 60 experiments with 70% training, 15% validation, and 15% testing. For DA, a regression analysis was used. The model was evaluated; it showed a 0.98147 correlation coefficient for training and 0.97247 in total data compared with 0.882 obtained by DA.

## 1. Introduction

A liposome is a vesicle frequently made of phospholipids and cholesterol [1]. Liposomes are used in different applications such as transfection [2], drug delivery [3], chemotherapy [4], cosmetics [5], and many others. Mechanical dispersion [6], solvents dispersion [7], and detergent removal [8] are different methods that exist to produce liposomes, but microfluidic micromixers were previously demonstrated to be robust and scalable methods of making size-controlled liposomes [9].

Liposome size (LZ) is an important factor in efficient cancer drug delivery [10]. That is why being able to determine the size of the liposomes before manufacturing them would allow for a lean process [11]. 

Data analysis (DA) tools were previously used to predict LZ fabricated by micromixers [9,12,13,14]. The models considered LZ as a dependent variable of the Flow Rate Ratio (FRR) and Total Flow Rate (TFR). The implementation of an artificial neural network (ANN) could allow for the development of more accurate models [15]. Artificial Intelligent strategies have become a common tool for pharmaceutical research [16]. ANN works as a “Universal algebraic function” that contemplates noise from experimental data [17], which can help predict the liposome size, finding patterns and relationships between the two data inputs. Comparative studies of both techniques are required to determine the one with the best performance [18]. 

Currently, there is no ANN-based model for the prediction of LZ; however, this technique was previously used successfully in the prediction of silver nanoparticles [19,20], droplet size in microfluidic devices [21], or the identification of operating parameters in microfluidic devices [22,23,24]. 

The Periodic Disturbance Mixer (PDM) is a micromixer designed for liposome production based on Dean forces. This device has a polynomial equation that estimates LZ using the FRR and TFR [14]. 

In this work, we compared the LZ prediction model based on a DA versus ANN two-layer feed-forward network, known as Fitnet [25], in a PDM when temperature, geometry, solvents, and lipids are constants, and the FRR and TFR are variables. We demonstrated that the ANN method had a higher regression number and a lower MSE than the DA model. 

## 2. Materials and Methods

### 2.1. Experimental Setup

To compare analyses between DA and ANN models successfully, the experimental setup used in “Surface Response Based Modeling of Liposome Characteristics in a Periodic Disturbance Mixer” was preserved. It consisted of PDM devices and three syringe pumps (2 Harvard Apparatus 11 plus 70-2212, 1 Norm-Ject 10 mL). One syringe pump is for the lipid-ethanol mixture; the second is for the MilliQ water; and the last one is for ethanol just for channel cleaning. Each syringe was connected using a 0.22 µm filter and tubing, as shown in Figure 1. The PDM was placed on the hot plate at 70 °C. The final mixture was collected, on the hot plate too, from the outlet in 4-mL scintillation vials prepared with MiliQ water for a final lipid concentration of 0.1 mg/mL for each sample. Samples were cooled down to room temperature, then stored at 4 °C.

The lipid mixture consisted of 1,2-dimyristoyl-sn-glycerol-3-phosphocholine (DMPC), cholesterol, and diacetyl phosphate (DHP) at a molar radio 5:4:1 and all diluted in ethanol with a final concentration of 5 mM.

### 2.2. Data Recollection 

The FRR and TFR were selected according to preceding work [14]. The first step was designed with Design of Experiment (DoE) and response surface methodology strategies, taking into consideration micromixer operation conditions and equipment restrictions. All the experiments were between 3 to 18 mL/h for the TFR and 1 to 12 mL/h for the FRR.

The size distribution by the intensity and polydispersity index (PDI) were measured by Dynamic Light Scattering (DLS) equipment, Zetasizer Nano S90 (Malvern, Worcestershire, United Kingdom). To obtain statistically significant results for liposome size, the LZ was obtained by the average of three independent measurement repetitions per sample. 

### 2.3. Prediction Models 

The liposome size prediction model based on DA techniques was previously supported [9,13]. It consisted of a reduced quadratic surface response model, Figure A1, based on the Central Composite Circumscribed Rotatable (CCCR) design, fitted using the TFR and FRR as independent variables of 29 samples. TFR^2^ and TFR*FRR were not considered because of non-probabilistic significance. The DA model is shown in Equation
(1)$$LZ=236.3-26.95FRR-4.437TFR+1.573{\left(FRR\right)}^{2}$$

For AI, a prediction model was used with the ANN, two-layer feed-forward network with sigmoid hidden neurons (hidden layer) and linear output neurons (output layer) (Figure 2). It consisted of 2 input and 1 output variables (Table 1), with 60 samples (Figure A2 and Figure A3). A total of 70% of data were used for training, 15% for validation, and finally 15% for testing. This ANN was programmed in MATLAB using the nnstart toolbox.

The flow chart of Figure 3 shows the process used to program the ANN prediction model; the first step consisted of importing input and output data in the workspaces; then, with the nftool command, the ANN algorithms (Fitnet) were accessed. The second step was to select the inputs and targets of the ANN from the workspace’s variables, and the third step was to select the percentage of data for training, validation, and test from the whole dataset. The fourth step was the determination of the neurons’ number in the hidden layer. Finally, the sixth step was the training where Levenberg–Marquardt was chosen as the training algorithm. If the regression coefficient was satisfactory, then the network parameters were stored, and if not, the ANN architecture was retrained.

## 3. Results and Discussions 

### 3.1. ANN Prediction Model

The LZ prediction model was developed in MATLAB with nftool. A heuristic approach was used to select the best training, validation, and testing parameters. The data used are shown in Table A1. A total of 42 data were employed for training, 9 for validation, and 9 for the test. All data were randomly selected.

The two layers of the ANN consisted of the hidden layer with 10 neurons and the output layer with a single neuron. The number of neurons in the hidden layer was selected based on the seed neurons’ number according to Equation (2) [16,29]:(2)$$N{}^{\circ}\;of\;neurons=\frac{\sqrt{1+8n}-1}{2}$$
where *n* is the number of samples.(3)$$N{}^{\circ}\;of\;neurons=\frac{\sqrt{1+8*60}-1}{2}=\frac{\sqrt{1+480}-1}{2}=\frac{\sqrt{481}-1}{2}=\frac{20.9}{2}=10.4$$

The training process was made with 10 neurons in the hidden layer. The ANN was retrained until it had a total regression number close to 1. 

Figure 4 is the MATLAB Training progress report. In this window, it is possible to know all of the ANN characteristics and performances. 

The performance analysis of the ANN is based on the correlation coefficient (R) and Mean Square Error (MSE). R is a statistic measurement of the relationship between variables and their association with each other and is given for the next Equation (4) [30].
(4)$$R=\frac{\sum \left(x_{i}-\bar{x}\right)\left(y_{i}-\bar{y}\right)}{\sqrt{\sum {\left(x_{i}-\bar{x}\right)}^{2}\sum {\left(y_{i}-\bar{y}\right)}^{2}}}$$
where *R* is the correlation coefficient, $x_{i}$ values of the x-variable in a sample, $\bar{x}$ mean of the values of the x-variables, $y_{i}$ values of the y-variable in a sample, $\bar{y}$ mean of the values of the x-variables. 

The *MSE* is used to determine how close a regression line is to the measured data. A *MSE* value close to 0 indicates that the model fits with the data [31]. Table 2 shows the performance analysis, and Figure 5 has the plots of the data and the model line.
(5)$$MSE=\frac{1}{Number\;of\;sample}\sum Square\;errors$$

The *square error* is defined in the next Equation (6).
(6)$$Square\;error={\left(Real\;value-forecast\right)}^{2}$$

Figure 6 is the histogram of the errors between target values and predicted values after training the ANN. On this graphic, the y-axis represents the number of samples from the dataset, and the x-axis is divided into 20 bins. The *width* of each bar represents the most common type of error, and it was calculated by the Equation (7), and in the case of our ANN was 4.2965 nm [19].
(7)$$Width=\frac{Maximum\;error-minimum\;error}{20}$$
(8)$$Width=\frac{34.06-\left(-51.87\right)}{20}=\frac{85.93}{20}=4.2965nm$$

The best validation performance was recorded during epoch number 8 (Figure 7. An epoch is referred to as one cycle over the complete training dataset [32]). In this case, Figure 8 demonstrates that errors are repeated three times after epoch number 8, and the process is stopped at epoch 11.

The code generated for the ANN is in Appendix C.

### 3.2. DA Prediction Model

According to Lopez [10], the performance of the LZ model was evaluated by R^2^, R^2^-adjusted, and R^2^-predicted. R^2^ values’ range is from 0 to 1, where 0 indicates that the model does not describe the process, and 1 shows that all data are on the regression line [21]. The adjusted R^2^ is a variant of R^2^ that has been adjusted for the number of forecasters in the model. R^2^-adjusted increases if the new term improves the model more than it would be expected by chance. It decreases when a forecaster improves the model by less than expected by chance. R^2^-predicted is used to indicate if the regression model predicts responses that have a good performance for new observations [33].

This model had R^2^ = 78.89%, R^2^-ajusted = 76.35%, and R^2^-predicted = 70.20%.

### 3.3. Comparation Models

The models were evaluated by R. For the DA model was used the R multiple (B1), and for ANN model all data R. Table 3 shows a better R for the ANN model with 0.97247 versus 0.7401 for DA model. 

### 3.4. Experimental Validation

After obtaining the two prediction models, five sets of the FRR and TFR were randomly selected for the experimental validation of the models, where the exclusion criterion was: having the same operation condition. The experimental corroboration samples were carried out with the same protocol for the collection of data from the models and characterized in the same equipment. The MSE was used to evaluate the performance of the two models.

Table 4 shows the MSE of external validation for the ANN and DA models. The square error was calculated for each measurement. The MSE was calculated with Equation (5), and the value obtained for the ANN model was 1.057, and 373.44 for DA model. 

According to Table 4, the MSE obtained with the eternal validation data was 373.44 for the DA model compared with 1.057 obtained for the ANN model. According to the MSE evaluation criteria, the ANN had a closer MSE to 0 with 1.057, meaning that the model fits better to the data than the DA model with 373.44. Therefore, this result is consistent with that obtained through R. 

## 4. Conclusions

This comparative study aimed to find out whether the DA or ANN was the most efficient method to estimate LZ. Previous research showed that ANNs were the most favorable option for predicting the size of silver nanoparticles. This study confirmed that the ANN was the better approach than the DA for predicting LZ.

The external validation data showed that the MSE in the ANN model was 1.057 contrasted with 373.44 obtained in the DA model. Using R, the DA model showed 0.8884 versus 0.97247 shown by the ANN model in all data.

This work is the first step to complete a universal model as it shows that the training of an ANN improves the regression coefficient compared to DA processes, which allows us to suppose that by expanding the number of variables that are involved in the generation of the model, it can improve its performance and also generalize it.

To generate a universal model, it is important to have an adequate database with the different micromixers currently designed, as well as the type of lipids, solvents, temperatures, flow rates, and mixing percentages to design a complex neural network that is capable of taking all the variables of the systems and thus be able to determine the size of the liposome generated with the given specifications. 

Currently, the use of micromixers for the mass production of liposomes has not been implemented. However, when contrasted with an equation that allows one to know the size of the liposome with the given conditions, it could help to promote this technology as one of the most viable by not requiring expensive laboratory equipment.

## Figures and Tables

**Figure 1 micromachines-12-01164-f001:**
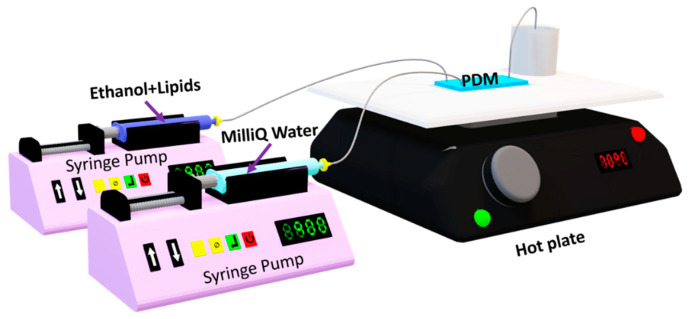
Experimental setup [26].

**Figure 2 micromachines-12-01164-f002:**
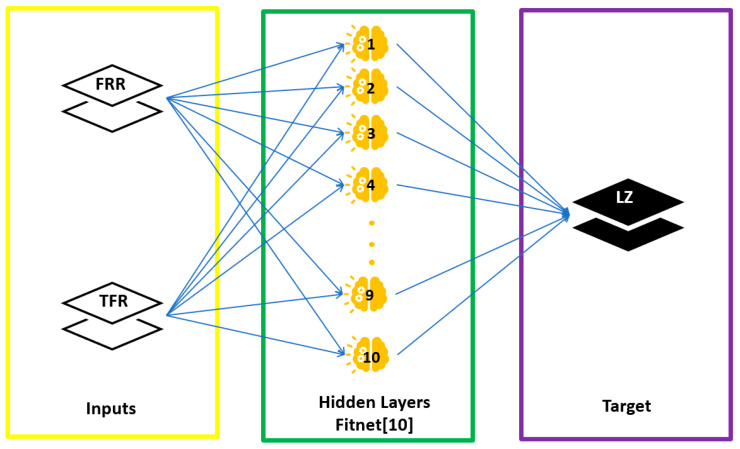
Schematic diagram demonstrating the model architecture of ANN prediction model.

**Figure 3 micromachines-12-01164-f003:**
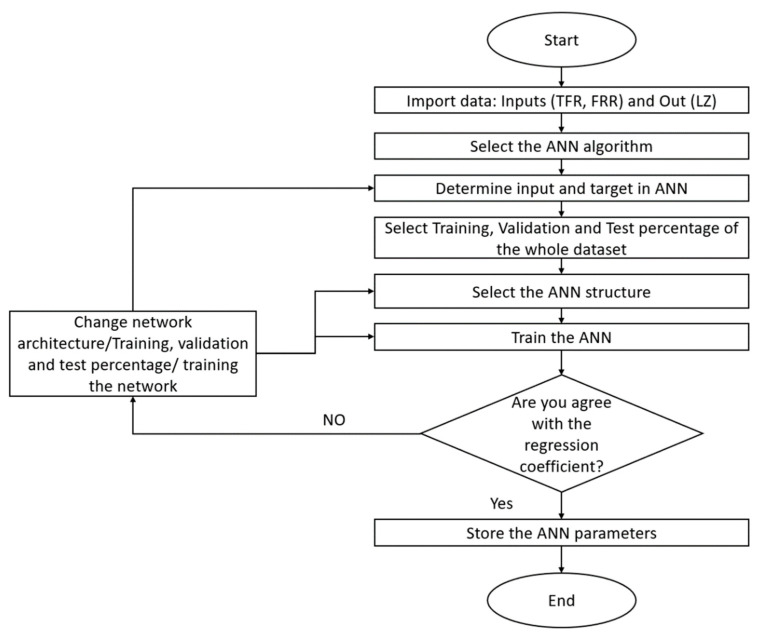
Flowchart used in MATLAB environment.

**Figure 4 micromachines-12-01164-f004:**
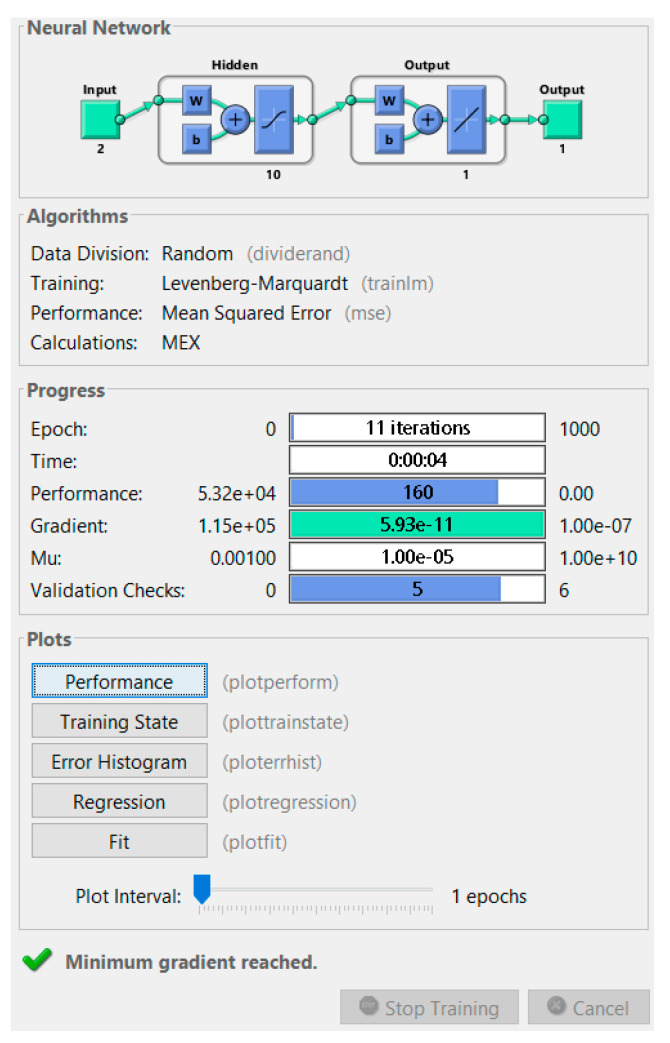
MATLAB Training progress report.

**Figure 5 micromachines-12-01164-f005:**
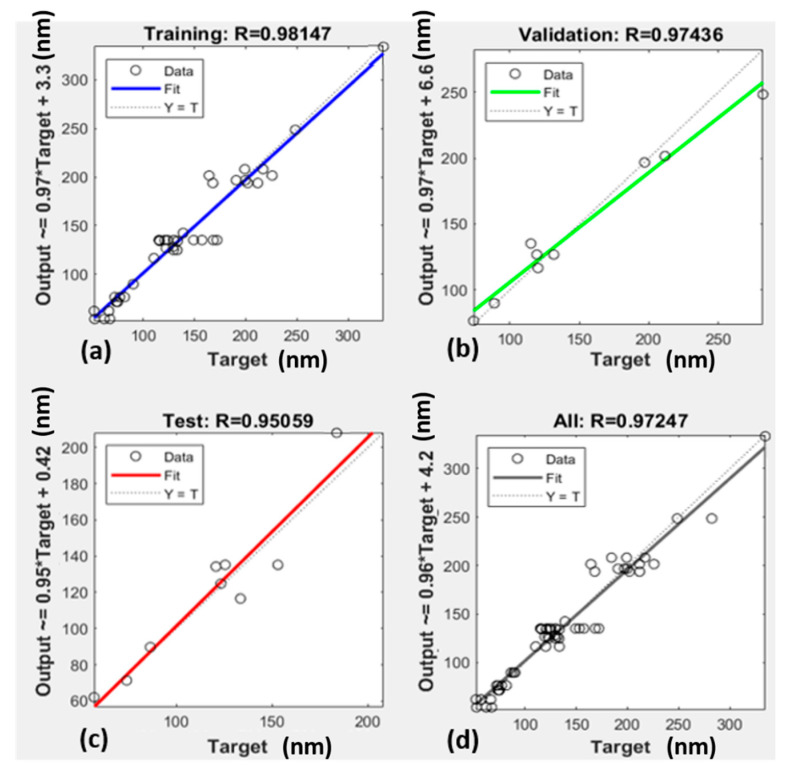
Linear Regression of ANN; (**a**) ANN training model Regression analysis for LZ; (**b**) ANN testing model Regression analysis for LZ; (**c**) ANN validation model Regression analysis for LZ; (**d**) ANN all data model Regression analysis for LZ.

**Figure 6 micromachines-12-01164-f006:**
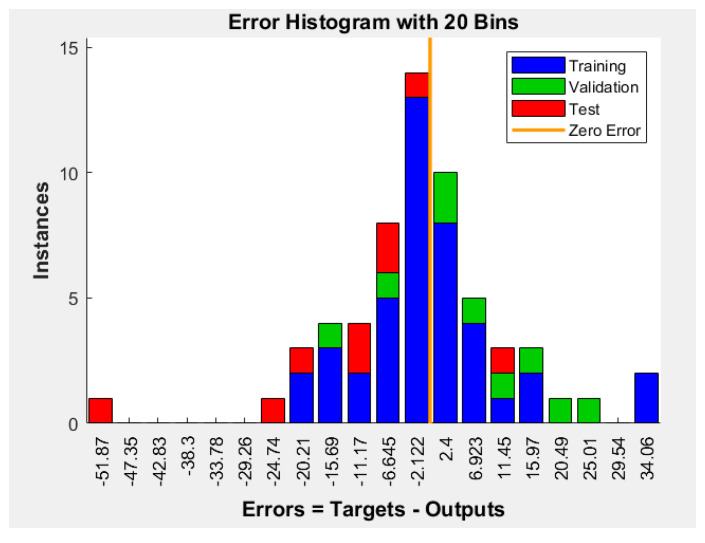
Error histogram with 20 bins plot.

**Figure 7 micromachines-12-01164-f007:**
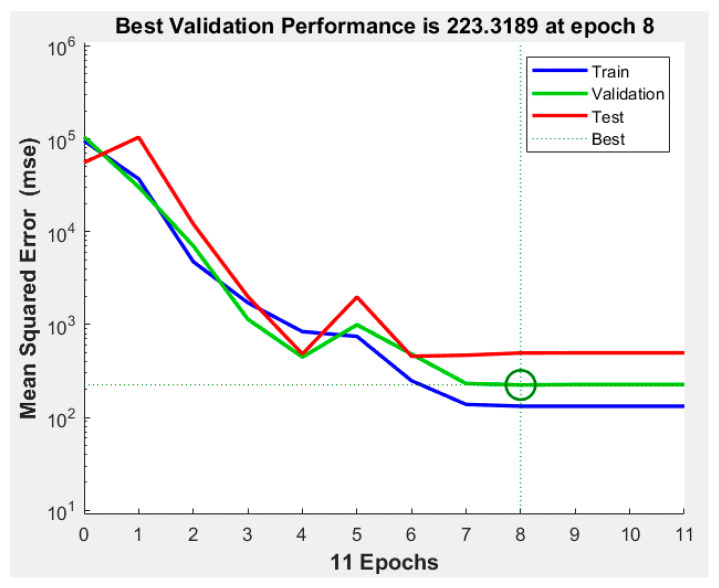
Best validation performance in ANN.

**Figure 8 micromachines-12-01164-f008:**
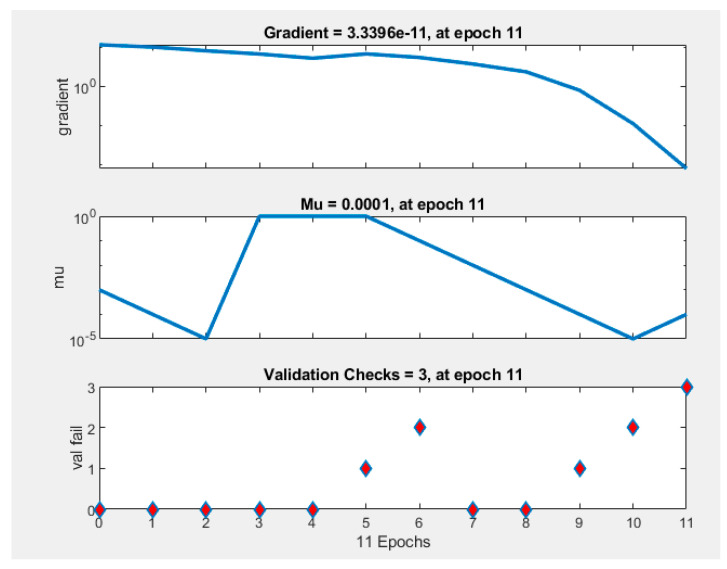
Training state performance plot. At the epoch 11 iteration, gradient is 3.3396 × 10^−11^, Mu = 0.0001 at epoch 11, and validation checks = 3 at epoch 11.

**Table 1 micromachines-12-01164-t001:** Variables of the ANN.

Variables	Units	Meaning
FRR (input)	-	The flow rate ratio is the fraction of flow between the water phase and solvent/lipid phase [27].
TFR (input)	ml/h	The total flow rate is the sum of flow between the water phase and solvent/lipid phase [28].
LZ (output)	nm	The liposome size is the average of three independent measurement repetitions of size distribution by intensity.

**Table 2 micromachines-12-01164-t002:** MSE and R coefficient for training, validation, and testing.

	MSE	R
Training	156.7893	0.98147
Validation	290.50693	0.97436
Testing	328.40462	0.95059
All	-	0.97247

**Table 3 micromachines-12-01164-t003:** Regression coefficient comparation between DA and ANN models.

Model	R
DA	0.8882
ANN	0.97247

**Table 4 micromachines-12-01164-t004:** Experimental validation performances using ANN and DA model.

Sample	Frr	Tfrr mL/h	Measurement LZ nm	ANN LZ nm	Square Error	DA LZ nm	Square Error
1	10.40	5.20	120.2	121.02	0.674	103.083	292.99
2	12.02	10.5	73.8	74.988	1.412	92.99	368.26
3	6.5	10.5	77.24	78.980	3.030	80.995	14.100
4	5	18.0	64.7	64.288	0.169	61.009	13.623
5	3.3	3.1	199.1	199.08	0.000	164.774	1178.27
				MSE	1.057	MSE	373.44

## Data Availability

https://doi.org/10.3390/Micromachines2021-09549.

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
