# Peer review of "Comparative Evaluation of Artificial Neural Networks and Data Analysis in Predicting Liposome Size in a Periodic Disturbance Micromixer"

_micromachines, 2021, doi:10.3390/mi12101164_

Round 1
Reviewer 1 Report
The manuscript is very well written. I congratulate the Authors. The manuscript can be accepted after minor revision.
- There are only two inputs and one output, why do we need to use ANN?
- What are the criteria for selecting training, validation, and testing data sets?
- ANN definitely outperforms Regression in many ways and claiming in the abstract again the same thing is not necessary.
- The selection of hidden neurons is based on reference no 18. if the number of hidden layers is more than one, what will be the number of neurons in the other hidden layers. Can we use the same formula?
- Equations 4, 5, and 6 are well-known to the modeling community, you can avoid these equations by giving appropriate references.
- The authors generated Figure 9 based on the data. They need to interpret the results with the mechanism existing in the system.
- The introduction part is not explaining the earlier research of ANN in the related fields. It is too short. The authors need to mention the intention of their work clearly. (Why?)
Reviewer 2 Report
Please find the document attached.

Author Response
Comment 1: The DA models were trained on 29 data-points, whereas ANNs were trained on 85% of the 60 data-points (51 data-points). Therefore, it is an unfair comparison. I suggest authors re-train the DA model using the updated 51 experimental data-points (85% of 60 data-points and leaving 15% for testing). Once new DA is trained it can be fairly compared to the ANNs.
Response: Thank you for this suggestion. It would have been interesting to explore this aspect. However, in the case of our study, we decided to use 29 data-points model because according with a two-variable Central Composite Circumscribed Rotatable (CCCR) 29 experiments are appropriate to calculate the coefficients of the second order polynomial regression model.
Comment 2: The authors should provide sufficient explanation on how the 5 external data-points were chosen and why there were not included in the original dataset. As the authors separate their data to train (70%), validate (15%), and test (15%) sets, there is no need to have an external dataset since it can skew the results and include selection biases. Specially, since the MSE of the ANNs for the test set is ~328 (Table 2) and the MSE for the external datapoints is only 1 (Table 4), while the MSE for DA for the same datapoints is 373 (Table 5). Therefore, it is highly likely that the low MSE (almost perfect) of the ANNs is due to the selection biases, therefore, I suggest including the 5 external data points in the training-testing process and rerunning the data selection multiple times and providing standard deviations so that the effect of selection biases can be better understood.
Below is an example of authors providing the accuracy results of ANNs for multiple runs and providing an average and standard deviation, as opposed to a singular value:
Response: Thank you for your comment, you have raised an important point here. We did a bibliographic review and found different models that used data analysis to predict the size of liposomes*manuscript references (9, 12-14). However, Liposome size is a non-linear process, and it is sensitive to disturbances. The regression coefficients reported were lower than 80%. We found that ANNs had been using by control engineering for identification a dynamic process with good results. This was the reason we opted to try this technique.
Comment 3: The introduction needs to be expanded upon to give a better idea of the state of the art in liposome production, needs for performance prediction, and machine learning in microfluidics.
Response: Thank you for pointing this out. We agree with this comment. Therefore, we modified the introduction in the lines 31-60, in the lines 33-36 explain the different methods to produces liposomes. Finally, in the lines 49-52 explain the machine learning in microfluidics.
“A liposome is a vesicle frequently made of phospholipids and cholesterol [1]. Liposomes are used in different applications such as transfection [2], drug delivery [3], chemotherapy [4], cosmetics [5], and many others. Mechanical dispersion[6], solvents dispersion [7], and detergent removal[8] are different methods that exist to produce liposomes, but microfluidic micromixers have been demonstrated to be robust and scalable methods of making size-controlled liposomes [9].
Liposome size (LZ) is an important factor in efficient cancer drugs delivery[10]. That is why being able to determine the size of the liposomes before manufacturing them would allow having a leaner process[11].
Data analysis (DA) tools have been used to predict LZ fabricated by micromixers [9, 12-14]. The models considered LZ as a dependent variable of Flow Rate Ratio (FRR) and Total Flow Rate (TFR). The implementation of an artificial neural network (ANN) could allow the development of more accurate models [15]. Artificial Intelligent strategies have become a common tool for pharmaceutical research [16]. ANN works like a “Universal algebraic function” that contemplates noise from experimental data [17], which can help predict the liposome size, finding patterns and relationships between the two data inputs. Comparative studies of both techniques are required to determine the one with the best performance [18].
Currently, there is no ANN-based model for the prediction of LZ, however, this technique has been used successfully in the prediction of silver nanoparticles [19, 20], droplet size in microfluidic devices [21], or the identification of operating parameters in microfluidic devices[22-24].
Periodic Disturbance Mixer (PDM) is a micromixer designed for liposome production based on Dean forces. This device has a polynomial equation that estimates LZ using FRR and TFR [14].
In this work, we compared the LZ prediction model based on DA versus ANN two-layer feed-forward network, known as Fitnet [25] in a PDM when temperature, geometry, solvents and lipids are constants, and FRR and TFR are variables. We demonstrated that the ANN method had a higher regression number and lower MSE than DA model. “
Comment 4: The authors provided MSE and R (or R2) for analyzing the accuracy of performance prediction. Providing other performance prediction metrics such RMSE (root mean square error), MAPE (mean absolute percentage error), and MAE (mean absolute error) can be very useful in terms of elucidating on what range of liposome sizes most errors occur or if there a lot of small errors or a few large errors. Therefore, I suggest adding these metrics to Table 2. For reference the authors can check the following publication:
Lashkaripour, Ali, Christopher Rodriguez, Noushin Mehdipour, Rizki Mardian, David McIntyre, Luis Ortiz, Joshua Campbell, and Douglas Densmore. "Machine learning enables design automation of microfluidic flow-focusing droplet generation." Nature communications 12, no. 1 (2021): 1-14.
Comment 5: Finally, I suggest authors adding a discussion on the generalizability of the ANN model to different microfluidic geometries or different fluid properties using Transfer Learning. As others mention briefly in the conclusion, it would be a great discussion point if authors provide their thoughts on what needs to be done to truly achieve Universal model for performance prediction. Additionally, it would be great if the authors provide additional discussion on how accurate performance prediction, enables additional features such as application-specific optimization and design automation. (Note to the authors: This comment doesn’t require any additional experiments and is solely a request for your thoughts on the matter).
Response: Thank you for pointing this out. We agree with this comment. Therefore, we modified the conclusion line 203-222.
This comparative study aimed to find out whether DA or ANN was the most efficient method to estimate LZ. Previous research showed that ANNs were the most favorable option for predicting the size of silver nanoparticles. This study confirmed that ANN was the better approach than DA for predicting LZ.
The external validation data showed that MSE in the ANN model was 1.057 contrasted with 373.44 obtained in the DA model. Unlike R, where the DA model showed 0.7401 versus 0.97247 shown by the ANN model in all data.
This work is the first step to complete a universal model as it shows that the training of an ANN improves the regression coefficient compared to AD processes, which allows us to suppose that by expanding the number of variables that are involved in the generation of the model, it can improve its performance and generalize it.
To generate a universal model, it is important to have an adequate database with the different micromixers currently designed, as well as the type of lipids, solvents, temperatures, flow rates, and mixing percentage to design a complex neural network that is capable to take all the values of the systems and thus be able to determine the size of the liposome generated with the given specifications.
Currently, the use of micromixers for the mass production of liposomes has not been implemented. However, when contrasted with an equation that allows knowing the size of the liposome with the given conditions, it could help to promote this technology as one of the most viable by not requiring expensive laboratory equipment.
Comment 6: What is the point of Figure 9, if Figure 9 is already published before and no side-by-side comparison is made to the ANN models? I would suggest either creating the same plot using ANNs and adding it to Figure 9 or moving Figure 9 to the supp info.
Response: Thank you for your comment. We moved Figure 9 at support information line 225
Comment 7: can also be represented in a figure for a side-by-side comparison of the two different predictive models. The extra data-points can be provided in the supplementary information as opposed to two full tables in the manuscript.
Response: Thank you for your comment.
Comment 8: Figure 5: The axes have no units.
Response: Thank you for your comment. We add the axes units to figure 5 line 147
Comment 9: Line 37: a dependent variable and flow rate àa dependent variable of flow rate
Response: Thank you for your comment. We modified the line 37, now Line 41
Comment 10: Line 38 TFR: Is typo, and should be total flow rate.
Response: Thank you for your comment. We corrected the typo in the line 38, now Line 42
Comment 11: Line 71: Flow rate ratio is given in units of ml/h, which is a typo.
Response: Thank you for your comment. We corrected the typo in the line 38, now Line 42
Reviewer 3 Report
Stiharu et al. report in this paper on the comparison between the artificial neural networks (ANN) and data analysis (DA) in order to predict the liposomes size, which is very interesting in the field of nanotechnology. The authors used MATLAB for the design of the ANN, whereas a regression analysis was employed for the DA. The authors conclude that the ANN is an approach better than the DA for predicting the liposomes size. Overall, the methodology and research work are OK and they reach the standard of the journal Micromachines. So that, my recommendation is to publish the work, but after the following revisions:
-
For the correlation coefficients [found in Abstract (line 24), Table 3, Conclusions (line 201), and Appendix B (Equation B1)], please add the initial zero, that is, change “.98147” by “0.98147” and so on.
-
The conclusions section should be more developed. Please, check this out.
-
The authors should increase the number of references, which results to be a bit low for this research field.
Author Response
Comment 1: For the correlation coefficients [found in Abstract (line 24), Table 3, Conclusions (line 201), and Appendix B (Equation B1)], please add the initial zero, that is, change “.98147” by “0.98147” and so on.
Response: Thank you for your comment. We added the initial zero in the next lines: 24,25,207, Table 3 and Equation B1
“…The model was evaluated; it showed a 0.98147 correlation coefficient for training and 0.97247 in total data comparing with 0.882 obtained by DA…”
“…0.8884 versus 0.97247 shown by the ANN model in all data…”
Comment 2: The conclusions section should be more developed. Please, check this out.
Response: Thank you for pointing this out. We agree with this comment. Therefore, we modified the conclusion line 203-222.
This comparative study aimed to find out whether DA or ANN was the most efficient method to estimate LZ. Previous research showed that ANNs were the most favorable option for predicting the size of silver nanoparticles. This study confirmed that ANN was the better approach than DA for predicting LZ.
The external validation data showed that MSE in the ANN model was 1.057 contrasted with 373.44 obtained in the DA model. Unlike R, where the DA model showed 0.7401 versus 0.97247 shown by the ANN model in all data.
This work is the first step to complete a universal model as it shows that the training of an ANN improves the regression coefficient compared to AD processes, which allows us to suppose that by expanding the number of variables that are involved in the generation of the model, it can improve its performance and generalize it.
To generate a universal model, it is important to have an adequate database with the different micromixers currently designed, as well as the type of lipids, solvents, temperatures, flow rates, and mixing percentage to design a complex neural network that is capable to take all the values of the systems and thus be able to determine the size of the liposome generated with the given specifications.
Currently, the use of micromixers for the mass production of liposomes has not been implemented. However, when contrasted with an equation that allows knowing the size of the liposome with the given conditions, it could help to promote this technology as one of the most viable by not requiring expensive laboratory equipment.
Comment 3: The authors should increase the number of references, which results to be a bit low for this research field.
Response: Thank you for pointing this out. We agree with this comment. Therefore, we increase of 23 to 34 references, lines 370-450
Round 2
Reviewer 2 Report
The authors made the effort to improve the paper. I suggest the paper to be accepted.